# Multi-Scale Crystal Plasticity Model of Creep Responses in Nickel-Based Superalloys

**DOI:** 10.3390/ma15134447

**Published:** 2022-06-24

**Authors:** Shahriyar Keshavarz, Carelyn E. Campbell, Andrew C. E. Reid

**Affiliations:** Materials Science Division, Thermodynamics and Kinetics Group, National Institute of Standards and Technology, Gaithersburg, MD 20899, USA; carelyn.campbell@nist.gov (C.E.C.); andrew.reid@nist.gov (A.C.E.R.)

**Keywords:** crystal plasticity, creep, nickel-based superalloys, morphology, composition

## Abstract

The current study focuses on the modeling of two-phase γ-$\gamma^{\prime }$ nickel-based superalloys, utilizing multi-scale approaches to simulate and predict the creep behaviors through crystal plasticity
finite element (CPFE) platforms. The multi-scale framework links two distinct levels of the spatial spectrum, namely, sub-grain and homogenized scales, capturing the complexity of the system responses as a function of a tractable set of geometric and physical parameters. The model considers two dominant features of $\gamma^{\prime }$ morphology and composition. The $\gamma^{\prime }$ morphology is simulated using
three parameters describing the average size, volume fraction, and shape. The sub-grain level is expressed by a size-dependent, dislocation density-based constitutive model in the CPFE framework with the explicit depiction of γ-$\gamma^{\prime }$ morphology as the building block of the homogenized scale. The
homogenized scale is developed as an activation energy-based crystal plasticity model reflecting intrinsic composition and morphology effects. The model incorporates the functional configuration of the constitutive parameters characterized over the sub-grain γ-$\gamma^{\prime }$ microstructural morphology. The developed homogenized model significantly expedites the computational processes due to the
nature of the parameterized representation of the dominant factors while retains reliable accuracy. Anti-Phase Boundary (APB) shearing and, glide-climb dislocation mechanisms are incorporated in the constitutive model which will become active based on the energies associated with the dislocations.
The homogenized constitutive model addresses the thermo-mechanical behavior of nickel-based superalloys for an extensive temperature domain and encompasses orientation dependence as well as the loading condition of tension-compression asymmetry aspects. The model is validated for diverse compositions, temperatures, and orientations based on previously reported data of single crystalline nickel-based superalloy.

## 1. Introduction

Nickel-based superalloys are widely utilized in the aerospace industry, especially in the hot sections of turbine engine components, such as blades and disks [1]. The corresponding superior mechanical properties, including exceptional yield stress and creep resistance at high-temperature, are attributed to the underlying γ-$\gamma^{\prime }$ microstructure characteristics [2]. The γ-matrix phase is predominantly composed of Ni, Cr, and Co having a face-centered cubic (FCC) lattice structure while the $\gamma^{\prime }\;$-precipitates phase, is a coherent, ordered ${\mathrm{Ni}}_{3}\mathrm{Al}$ intermetallic with an $L1_{2}$ crystal structure.

There have been significant investments to enhance the efficiency of turbine engines, consistently leading to considerable demands for the materials used in the hot sections. The trend has been towards higher melting points and longer service lifetimes, entailing improved creep response and fatigue resistance. Consequently, several computational approaches along with multiple generations of these alloys with minor compositional modifications have been established and designed over time. Accordingly, elements such as Mo, W, Re, etc. with various concentrations have been added to these superalloys and empirically tested for their effects on the creep properties. The primary reasons for pursuing experiential approaches are the complex dislocation mechanisms and stress states associated with the hot sections in turbine engines, which require precise, physics-based models; hence, intricate constitutive relations to acquire powerful means for analysis and design of these parts accurately, without incurring excessive computational time and cost.

Surmounting these impediments, the objective of this work has been to attain a comprehensive, yet, viable solution with regard to processing time and resources. Thereby, constitutive models are developed embodying the dominant factors with the presiding effects on the creep properties of Ni-based superalloys. The morphology of the microstructure, accounted as of the controlling characteristics, depends on the cooling rate [3,4], as well as internal stresses [5] during heat-treatment processes. The intrinsic dislocation mechanisms are largely affected by the morphology of a microstructure where the precipitates act as obstacles against the further motion of dislocations, either looping around or shearing them, depending on the temperature and stress level. The volume fraction of $\gamma^{\prime }\;$phase is another microstructural criterion with substantial effects on the mechanical properties of superalloys [6,7].

The deformation behavior under extreme loading and temperature conditions has been studied extensively, both for single-crystal [8,9] and polycrystalline [10,11] Ni-based superalloys. Considering the broad range of service temperatures, the relevant dislocation mechanisms governing the associated responses differ as temperature changes. At lower temperatures, the activation of octahedral slip systems is mainly reported in both phases. As temperature increases, anomalies in mechanical properties are observed attributed to the dislocation mechanisms in the intermetallic $\gamma^{\prime }\;$phase where screw dislocations tend to be locked in a Kear-Wilsdorf (KW) configuration due to cross-slip. The rate of lock formation rapidly expands as temperature elevates to around 725 °C. At lower temperatures, any possible creep deformation is governed by dislocation-based shearing processes, which is the case in excessive loading conditions, whereas, at higher temperatures, T>800 °C, dislocation climb mechanisms are the dominant component in creep deformation [7]. This work focuses on dislocation-based shearing and climb mechanisms The rafting processes dominates at higher temperatures [12,13] as the gamma prime precipitates begin to coarsen. Thus the current model predictions begin to diverge from the experimental results above 900 °C.

Taking all these effective parameters and mechanisms into account along with alleviating the computational encumbrances, a multi-scale method with the characteristics of composition and morphology variables along with underlying dislocation mechanisms is sought.

Crystal plasticity finite element models (CPFEM) [14,15] are employed to collect the data hierarchically at each scale using proper constitutive models in which they can include microstructural property information. Meso-scale computations of superalloys, including both γ-$\gamma^{\prime }\;$phases at single crystals, as well as the grain information at polycrystalline levels, have been conducted using phenomenological constitutive laws in [16,17]. Constitutive hardening parameters are presented in terms of some functional forms of the average precipitate size in most of the models. Analytical models have been proposed using simplifying assumptions in terms of dislocation distributions under uniaxial and monotonic loads [18]. CPFEM with implicit dependencies on the precipitate size, and volume fraction at the single crystal and grain size at the polycrystalline scales are propose for a random distribution of precipitates in [19]. Physics-based models such as dislocation-density-based crystal plasticity models of creep and fatigue are postulated in a multi-scale framework [20,21], where the mechanical property dependencies on microstructural characteristics like average $\gamma^{\prime }\;$precipitate size and volume fraction are accommodated by parameters obtained through fitting experimental data. In the current work, the microstructural details for a broad temperature range with diverse dislocation mechanisms are utilized hierarchically enabling the simulation of mechanical responses as the functions of morphology and composition including stress-strain curves and creep behaviors at different temperatures, loading conditions; tension and compression as well as crystal orientations. The idea is aimed to be a beneficial approach for design processes.

In consequence, the system is divided into two length scales: (1) the sub-grain level characterized by the morphology of the $\gamma^{\prime }\;$phase with the explicit representation of γ-$\gamma^{\prime }\;$phases and (2) the homogenized scale with the implicit effects of morphology through parametric constitutive relations. In the sub-grain extent, a physics-based creep constitutive description is developed containing non-Schmid components with dislocation densities as the prime state variables. On the homogenized level, the constitutive model involves a non-Schmid structure; however, with inherent effects of morphology as the normative parameters. This model is able to capture temperature variation from room temperature to 900 °C for distinctive crystal orientations while retaining its sensitivity to tension and compression loading conditions. Crystal plasticity finite element methods are utilized to hierarchically incorporate information at each spatial spectrum and to represent size and morphology effects based on a multifunctional approach to simulate and predict the creep responses of Ni-based superalloys in extreme temperature environments. The current study delineates the results for the single-crystal of Ni-based superalloys; however, the method can be applied to polycrystalline superalloys as well.

## 2. Materials and Methods

An advanced multi-scale approach is introduced to expedite computational procedures while retaining acceptable precision. In this respect, the theoretical basis of entropic kinetics, plastic deformation kinematics, and statistical mechanics of a concomitant system at sub-grain and homogenized levels are utilized and adjusted to acquire compatible constitutive models embodying the dominant dislocation mechanisms and the salient features of size, volume fraction, and shape of the precipitates. Subsequently, the parameters in the developed constitutive models are calibrated with experimental data. The monads of the multi-scale framework are investigated by CPFE methods in the large deformation platform to model and predict the thermomechanical responses of nickel-based superalloys. This work is aimed at analyzing creep responses using a homogenized grain-scale crystal plasticity model where the functional hardening parameters in terms of $\gamma^{\prime }\;$morphology are introduced. The general scope of the multi-scale model, beginning from the sub-grain γ-$\gamma^{\prime }\;$to the homogenized level, is illustrated in Figure 1.

The objective of this multi-scale scheme is the behavioral analysis and prediction of single-crystalline superalloys. This procedure with latent precipitate effects in the homogenized microstructure allows proceeding with tractable computational processes. In order to meet the target, the coarse-graining must be initiated from a single crystal as a representative cell for turbine blades demonstrated in Figure 1a. Hence, a sequence of steps is followed to generate a hierarchical framework which originally starts with the development of a crystal plasticity finite element model of the smallest scale by the explicit representation of sub-grain level using representative volume element. To capture the size and morphology effects, a dislocation-density-based crystal plasticity model is utilized [22]. At the next step, the development of an activation-energy-based crystal plasticity model is pursued at the single crystal context utilizing homogenization as the connecting tool. The homogenized model incorporates the effect of the morphology through functional forms of the hardening parameters that resemble the implicit effects of the precipitates in the constitutive model. The resulting multi-scale model has the potential of significantly expediting the crystal plasticity simulations while retaining sufficient accuracy. The fundamental physics and rationales behind the development of the constitutive models at both sub-grain and homogenized scales are described in the subsequent sections.

### 2.1. Plastic Deformation Kinematics

Based on the finite strain kinematics, the total deformation gradient, F, is decomposed into an elastic part, $F^{e}$, for elastic stretch and rigid-body rotations and a plastic part, $F^{p}$, without volume change, associated with slip in the absence of rotation, in a multiplicative form of $F=F^{e}F^{p}$. The velocity gradient related to the plastic part, $L^{p}$, can be calculated as a function of the rate of plastic shear strain, $\dot{\gamma^{\alpha }}$, on all slip systems, α, as
(1)$$\begin{array}{cc} L^{p} & ={\dot{F}}^{p}F^{-p}=\sum_{\alpha =1}^{N}{\dot{\gamma }}^{\alpha }s{}_{0}^{\alpha }=\sum_{\alpha =1}^{N}{\dot{\gamma }}^{\alpha }m{}_{0}^{\alpha }⊗n{}_{0}^{\alpha }\;, \end{array}$$
where $s{}_{0}^{\alpha }=m{}_{0}^{\alpha }⊗n{}_{0}^{\alpha }$ is the Schmid tensor for the slip direction, $m{}_{0}^{\alpha }$, and slip plane normal, $n{}_{0}^{\alpha }$, in the reference configuration. The evolution of the deformation gradient corresponding to the plastic part, ${\dot{F}}^{p}$, is calculated from this equation. Thus, an integration strategy is used to calculate the total plastic part of the deformation gradient, and thereafter, the elastic part of the deformation through the multiplicative decomposition of F. The Green-Lagrange strain tensor and second Piola-Kirchhoff stress are, respectively, evaluated in the intermediate configuration with
(2)$$\begin{array}{cc} E^{e}=\frac{1}{2}\left(F^{e^{T}}F^{e}-I\right)\;\;\;\mathrm{and}\;\;\;\;S & =\det\left(F^{e}\right)F^{e^{-1}}\sigma F^{e^{-T}}=C:E^{e}, \end{array}$$
where I is the identity tensor, σ the Cauchy stress tensor, and C the fourth order anisotropic elasticity tensor.

### 2.2. Entropic Kinetics and Constitutive Model at the Sub-Grain Scale

Mechanical properties of alloys, including nickel-based superalloys in extreme conditions, are predominantly undertaken through a system in a high probability condition arising from enhanced entropy states. Considering the reference rate of change of entropy, $\dot{S}=\frac{DS}{Dt}$, with position vector, *r*, and time, *t*, total entropy generation, $E_{s}\left(t\right)$, is described with [23]
(3)$$\begin{array}{cc} E_{s}\left(t\right) & =\frac{D}{Dt}\int_{\Omega }S_{v}\left(r,t\right)\;dv+\int_{\partial \Omega }\frac{F_{T}\left(r,t\right)}{T(\lambda ,t)}\cdot n\;ds-\int_{\Omega }\frac{S_{O}\left(r,t\right)}{T(\lambda ,t)}\;dv\;, \end{array}$$
where the body is assumed to have volume Ω and surface ∂Ω in quasi-static conditions at all configurations. $S_{v}\left(r,t\right)$ is the entropy per unit volume, $F_{T}\left(r,t\right)$ the material heat flux per unit surface, $S_{O}\left(r,t\right)$ the entropy source per unit time and volume, n the normal boundary surface vector, and T(r,t) the thermal scalar field. The plastic properties of these alloys give rise to positive localized entropy production mainly as a function of internal variables involved in thermodynamical procedures. Consequently, the Helmholtz free-energy function, $H(S,T\left(r,t\right),\lambda_{x})$, is defined as a function of the second Piola-Kirchhoff stress, thermal field, and internal variables, respectively, where, $\lambda_{x}$ represents plastic deformation mechanisms in accordance with any admissible inelastic experiences, x=1,2,...,n. Internal entropy in nickel-based superalloys can be generalized to a variable composition of point, line, and volume defects and mechanisms that represents dislocation velocity as a dominant factor to be considered for obtaining a proper constitutive relation. In this respect, the average velocity of dislocations between barriers with a specific mean free path, $m_{d}$, can be written in terms of the Gibbs free energy, ΔG, and Boltzmann constant, $K_{B}$, as $v^{\alpha }=\frac{m_{d}}{t_{d}}\left[\exp\left(-\frac{\Delta G}{K_{B}T}\right)\right]$[24] when the running time, $t_{d}\left(\tau ,T\right)$, is a function of applied stress, τ, and thermal fluctuations. This relation can be sufficiently expressed based on internal and external dissipative variables accordingly entailing the Helmholtz free energy along the process by
(4)$$\begin{array}{cc} \; & \Delta G=\Delta H(\tau_{a}^{i},\tau_{r}^{j})\;, \end{array}$$
as a nonlinear function of applied, $\tau_{a}$, and resistance, $\tau_{r}$, stresses with allocated exponents of *i* and *j*.

The constitutive model at the sub-grain scale is inspired by [25] where the plastic shear strain rate on a slip system follows the Orowan equation, ${\dot{\gamma }}^{\alpha }=\rho {}_{m}^{\alpha }b^{\alpha }v^{\alpha }$, as a function of the mobile dislocation density, $\rho {}_{m}^{\alpha }$, and Burgers vector, $b^{\alpha }$. The parallel and perpendicular slip resistances, $\tau {}_{pass}^{\alpha }$ and $\tau {}_{cut}^{\alpha }$, respectively, influence the average velocity of the dislocations which can be represented by
(5)$$\begin{array}{cc} v^{\alpha } & =v_{0}\exp\left(-\frac{Q}{K_{B}T}\right)\sin\;h{\left(\frac{|\tau^{\alpha }|-\tau {}_{pass}^{\alpha }}{\tau {}_{cut}^{\alpha }}\right)}^{p}\mathrm{sign}\left(\tau^{\alpha }\right)\;. \end{array}$$

The slip system dependent hardening parameters are considered [26] as
(6)$$\begin{array}{cc} \tau {}_{pass}^{\alpha } & =c_{1}Gb{}^{\alpha }\sqrt{\rho {}_{P}^{\alpha }+\rho {}_{F}^{\alpha }}\;\;\;,\;\;\;\tau {}_{cut}^{\alpha }=\frac{c_{2}K_{B}T}{b^{2}}\sqrt{\rho {}_{F}^{\alpha }}\;, \end{array}$$
where *G* is the shear modulus and $c_{1}$ and $c_{2}$ are calibrated material constants as 4 and 0.3 respectively [27]. The passing and cutting hardening parameters are the functions of the parallel, $\rho_{P}$, and forest, $\rho_{F}$, dislocation densities which are projected dislocation densities aligned and orthogonal to the slip planes. The sources of hardening are statistically stored dislocations (SSDs) that are scalar and rate-dependent accounted for dislocation generation and annihilation processes. However, due to the existing gradient of plastic deformation, especially at the interface of γ-$\gamma^{\prime }\;$where the misfit is of concern, there will be another source contributing to the slip resistance from geometrically necessary dislocations (GNDs) [28]. GNDs are introduced to consider this phenomenon that is more pronounced in the γ-$\gamma^{\prime }\;$ boundaries. The strain gradient effects in these superalloys due to the dimensions of $\gamma^{\prime }\;$are captured by GNDs through the morphological parameters of shape, average size, and volume fraction of $\gamma^{\prime }\;$. The higher volume fraction for the same average size results in more precipitates within a unit cell that creates larger gradient of plastic deformation. Whereas, the larger average size for the same volume fraction increases the channel width; hence, lowers the gradient of plastic deformation. Thereby, the parallel and forest dislocation densities can be stated in terms of a scalar SSDs and vectorial form of GNDs [26,29], where the mobile dislocation density $\rho {}_{m}^{\alpha }$ is expressed [27] with
(7)$$\begin{array}{cc} \rho {}_{m}^{\alpha } & =\frac{c_{3}K_{B}T\sqrt{\rho {}_{F}^{\alpha }\rho {}_{P}^{\alpha }}}{Gb^{3}}\;. \end{array}$$
where $c_{3}$ can be evaluated from $c_{1}$ and $c_{2}$ with $c_{3}=\frac{2c_{2}}{c_{1}}$.

### 2.3. Constitutive Model at the Homogenized Scale

Dislocation mechanisms of single-crystalline nickel-based superalloys are commonly investigated in both γand $\gamma^{\prime }\;$phases [9]. At lower temperatures, octahedral slip systems are active where glide occurs on these slip systems in both phases as exhibited in Figure 2. In the case of sufficient stress, both phases can be sheared according to Figure 3a; however, the nature of glissile dislocations is different in γand $\gamma^{\prime }\;$. As observed in Figure 2a, γphase acts as a regular FCC crystal structure having a dislocation with $\frac{1}{2}\langle 110\rangle$ in length where the span of a full dislocation in $\gamma^{\prime }\;$phase, called a super-dislocation, is almost twice as $\langle 110\rangle$, Figure 2b.

Super-dislocations cover a broad temperature range and are formed through the dissociation of a screw super-dislocation into two super-partials with a Burgers vector of $\frac{1}{2}\langle 1\bar{1}0\rangle$. This process creates a planar fault, anti-phase boundary, or APB within the $\gamma^{\prime }\;$. Afterward, these super-partials basically split into two Shockley partials to form a complex stacking fault (CSF) with Burgers vectors of $\frac{1}{6}\langle 11\bar{2}\rangle$. As the temperature rises, the screw dislocations in the $\gamma^{\prime }\;$phase can change slip planes and lock in a KW configuration due to cross-slip [30]. The rate of lock formation intensifies as temperature increases to approximately 725 °C. The locking mechanism entails the activation of the cube slip systems in addition to the octahedral ones. This process is further discernible for temperatures above the peak cross-slip rate or critical temperature of around 725 °C. Nevertheless, above the critical temperature, edge and screw dislocations on cube planes glide without any cross-slip. Consequently, the cross-slip mechanism is appended to the crystal plasticity constitutive model by considering non-Schmid components of the shear stress [25] where the constitutive model accounts for tension-compression asymmetry of different crystal orientations.

At high temperatures with lower applied stress, due to the lack of enough energy, dislocations climb along the interface of γ-$\gamma^{\prime }\;$towards the matrix where glide takes place in the channel prior to the next precipitates [31] as shown in Figure 3b.

To incorporate the climb-glide dislocation mechanism into the crystal plasticity, the Orowan relation is modified to ${\dot{\gamma }}^{\alpha }=\xi \rho {}_{g}^{\alpha }b^{\alpha }v^{\alpha }$ merely for the glide phenomenon since the precipitates are not sheared and plastic deformation passes off inside the γphase. Here, $\xi =1-V_{p}$, where $V_{p}$ represents the $\gamma^{\prime }\;$volume fraction. The mobile dislocation density is calculated in terms of glide and climb dislocation densities, respectively, as $\rho {}_{m}^{\alpha }=\rho {}_{g}^{\alpha }+\rho {}_{c}^{\alpha }$. The evolution of the glide dislocation density is estimated [31] by
(8)$$\begin{array}{cc} \; & {\dot{\rho }}_{g}^{\alpha }=\rho_{c}^{\alpha }V_{p}\frac{2b}{d}\Gamma_{e}^{\alpha }-\rho_{g}^{\alpha }\frac{v^{\alpha }}{w}\;, \end{array}$$
where *d* and *w* are the average size and spacing between the precipitates as denoted in Figure 3b and $\Gamma_{e}^{\alpha }$ is the released frequency for a given slip system. By rewriting $\rho_{c}^{\alpha }$ in terms of mobile and glide dislocation densities and taking into account that the time required for a dislocation to glide from one precipitate to another is much shorter than releasing a trapped dislocation, Equation (Equation 8) is reformed to
(9)$$\begin{array}{cc} \; & {\dot{\rho }}_{g}^{\alpha }+\rho_{g}^{\alpha }\frac{v^{\alpha }}{w}=\rho_{m}^{\alpha }v_{p}\frac{2b^{\alpha }}{d}\Gamma_{e}^{\alpha }\;. \end{array}$$

The glide dislocation density is obtained by solving this equation in terms of the mobile dislocation density as
(10)$$\begin{array}{cc} \; & \rho_{g}^{\alpha }=\rho_{m}^{\alpha }V_{p}\frac{2\Gamma_{e}^{\alpha }}{v^{\alpha }d}w\;. \end{array}$$

At this stage, the plastic shear strain rate for a given slip system is represented by
(11)$$\begin{array}{cc} \; & {\dot{\gamma }}^{\alpha }=2\rho_{m}^{\alpha }V_{p}\left(1-V_{p}\right)b^{\alpha }\frac{w}{d}\Gamma_{e}^{\alpha }\;, \end{array}$$
and escape frequency is achieved through a vacancy-emitting mechanism with regard to the Gibbs free energy [32] with
(12)$$\begin{array}{cc} \; & \Gamma_{e}^{\alpha }={\dot{\gamma }}_{0}^{\alpha }\exp\left\{-\frac{\Delta G^{\alpha }}{K_{B}T}\right\}\;, \end{array}$$
or according to [22] by
(13)$$\begin{array}{cc} \; & \Gamma_{e}^{\alpha }={\dot{\gamma }}_{0}^{\alpha }\exp\left\{-\frac{Q}{K_{B}T}{\left[1-{\left(\frac{\tau_{eff}^{\alpha }}{s^{\alpha }}\right)}^{p}\right]}^{q}\right\}\;, \end{array}$$
where the effective shear stress is determined as the difference between the resolved shear stress and passing stress, $\tau_{eff}^{\alpha }=\tau^{\alpha }-\tau_{pass}^{\alpha }$, respectively. The total slip resistance due to cutting and cross-slip is $s^{\alpha }=\tau_{cut}^{\alpha }+\tau_{cross}^{\alpha }$ [25]. The first part in Equation (Equation 13) is the effective diffusivity for the climb since the mass transport is affected by the vacancy flow [32] for which
(14)$$\begin{array}{cc} \; & D_{eff}^{\alpha }={\dot{\gamma }}_{0}^{\alpha }\exp\left\{-\frac{Q^{\alpha }}{K_{B}T}\right\}\;. \end{array}$$

Thereby, the plastic shear strain rate for a given slip system is stated as
(15)$$\begin{array}{cc} \; & {\dot{\gamma }}^{\alpha }=2\rho_{m}^{\alpha }V_{p}\left(1-V_{p}\right)D_{eff}^{\alpha }\left(\frac{w}{b^{\alpha }d}\right)\exp\left[1-\left(\frac{\tau_{eff}^{\alpha }}{\tau_{cut}^{\alpha }+\tau_{cross}^{\alpha }}\right)\right]\;. \end{array}$$

Accordingly, the evolution laws for the passing and cutting stresses [25] are, respectively, obtained through
(16)$$\begin{array}{cc} \; & {\dot{\tau }}_{pass}^{\alpha }=\sum_{\beta =1}^{n}h^{\alpha \beta }|{\dot{\gamma }}^{\alpha }\sin\left(n^{\alpha },t^{\beta }\right)|\;\;\;,\;\;\;{\dot{\tau }}_{cut}^{\alpha }=\sum_{\beta =1}^{n}h^{\alpha \beta }|{\dot{\gamma }}^{\alpha }\cos\left(n^{\alpha },t^{\beta }\right)|\;\;, \end{array}$$
where $h^{\alpha \beta }=q^{\alpha \beta }\left[h_{0}{\left(1-\frac{\tau_{t}^{\alpha }}{\tau_{sat}^{\alpha }}\right)}^{r}\right]$ is the hardening parameter with ${\dot{\tau }}_{t}^{\alpha }=\sqrt{{\left({\dot{\tau }}_{pass}^{\alpha }\right)}^{2}+{\left({\dot{\tau }}_{cut}^{\alpha }\right)}^{2}}$ as the rate of the total shear resistance and $q^{\alpha \beta }=q+\left(1-q\right)\delta^{\alpha \beta }$ with $q^{\alpha \alpha }=1$ and $q^{\alpha \beta }=1.4$. In addition, the cross-slip resistances are different for octahedral and cube slip systems since the former is a function of non-Schmid components, temperature, and APB energies whereas the latter is just a function of temperature [25] which is outlined by
(17)$$\begin{array}{ccc} \tau {}_{cross}^{\alpha }=\; & \left\{\begin{array}{cc} \tau {}_{co}^{\alpha }=\tau {}_{co}^{\alpha }\left(\tau {}_{pe}^{\alpha },\tau {}_{se}^{\alpha },\tau {}_{cb}^{\alpha },T,\Gamma_{111},\Gamma_{010}\right) & \mathrm{on}\;\mathrm{octahedral}\;\mathrm{slip}\;\mathrm{systems}\\ \tau_{cc}=\tau_{cc}\left(T\right) & \mathrm{on}\;\mathrm{cube}\;\mathrm{slip}\;\mathrm{systems} \end{array}\right. & \end{array}$$

The last unknown variable in the Orowan equation is the mobile dislocation density. A simple approach to obtain this term is by the given external stress that generates the maximum plastic deformation, or the given plastic deformation rate that causes minimum external stress; thus
(18)$$\begin{array}{cc} \; & \frac{\partial \tau^{\alpha }}{\partial \rho_{m}^{\alpha }}=0\;. \end{array}$$

Rearranging Equation (Equation 15) in terms of resolved shear stress, $\tau^{\alpha }$, and minimizing it with respect to the mobile dislocation density result in
(19)$$\begin{array}{cc} \; & \rho_{m}^{\alpha }=\frac{c_{1}}{G^{2}{b^{\alpha }}^{2}}\tau_{pass}^{\alpha }\left(\tau_{cut}^{\alpha }+\tau_{cross}^{\alpha }\right)\;, \end{array}$$
where $c_{1}$ is a dislocation mobility constant. To estimate the effective diffusivity, Arrhenius representation of $D_{eff}^{\alpha }=D_{eff0}^{\alpha }\exp\left(-\frac{Q_{eff}^{\alpha }}{K_{B}T}\right)$ is evaluated where $D_{eff0}^{\alpha }$ and $Q_{eff}^{\alpha }$ are the effective pre-exponential coefficient and activation energy, respectively.

The effective activation energy can be evaluated in terms of the activation energy [31] of the constituents in the composition assuming that the effect of solute elements on the solvent changes the effective activation energy in pure nickel by average weighted percentage from the different solutes, that is
(20)$$\begin{array}{cc} \; & Q_{eff}^{\alpha }=Q_{Ni}^{\alpha }+\sum_{m}x_{m}Q_{m,Ni}^{\alpha }\;, \end{array}$$
where $Q_{Ni}^{\alpha }$ is the activation energy for self-diffusion of nickel and $Q_{m,Ni}^{\alpha }$ the activation energy for the solute with the atomic concentration of *m*. Different methods are reported to evaluate the effective pre-exponential coefficients [31], among which, the arithmetic method is chosen in the current study to be more compatible with the rest of computational structure. This method represents $D_{eff0a}^{\alpha }=\sum_{m}x_{m}D_{0m,Ni}^{\alpha }$, where $D_{0m,Ni}^{\alpha }$ is the pre-exponential factor of solute concentration in nickel. Ultimately, by incorporating the climb-glide mechanism, the constitutive model is stated as
(21)$$\begin{array}{ccc} \; & {\dot{\gamma }}^{\alpha }=2\rho_{m}^{\alpha }V_{p}\left(1-V_{p}\right)\left[D_{eff0}^{\alpha }\exp\left(-\frac{Q_{eff}^{\alpha }}{K_{B}T}\right)\right]\left(\frac{w}{b^{\alpha }d}\right)\exp\left[1-\left(\frac{\tau_{eff}^{\alpha }}{\tau_{cut}^{\alpha }+\tau_{cross}^{\alpha }}\right)\right]\;. & \; \end{array}$$

The computational scheme to capture creep responses in the CPFE framework is demonstrated in Figure 4 where the homogenized constitutive model is utilized to capture the effects of the morphology and composition in nickel-based superalloys.

#### Calibration and Effective Parameters

The parameters to be calibrated for both constitutive models along with the functional constants associated with the homogenized model are stated in the reference [25] in Equation (16) and Tables 1 and 2.

## 3. Results and Discussions

### 3.1. Arithmetic Method to Calculate Diffusion Coefficients

To obtain the effective diffusion coefficient, $D_{eff0}^{\alpha }$, four different generations of single-crystal nickel-based superalloys in terms of compositions are selected as demonstrated in Table 1.

The effective diffusion coefficients and activation energy for these four assemblies are calculated from the data in [31,33] as indicated in Table 2.

The results in accordance with the considered information and data are plotted with the format of plastic strain versus time and illustrated in Figure 5. The dislocation mobility $c_{1}$ in Equation (Equation 19) with the arithmetic method is $3.5\;\times \;10^{-6}$. Simulations are performed for a microstructure with precipitates having cuboidal shapes, 70% volume fraction and 450 nm average size for SRR99 [34], 61% volume fraction and 490 nm average size for RENE-N5 [35], 68% volume fraction and 400 nm average size for CMSX-4 [36], and 63% volume fraction and 350 nm average size for TMS-75 [37]. The applied load is 400 MPa at 900 °C.

### 3.2. Homogenized Constitutive Model Validations

In this part, three distinct sets of simulations are performed to investigate the effects of temperature, load intensity, and orientation on the model towards validating the predictive characteristics of the constitutive model.

#### 3.2.1. Composition Effects on Creep Curves

The main feature of the developed constitutive model is reflecting the nominal composition effects. To examine this sensitivity, four different compositions are tested and the results are compared against the experimental data. The nominal chemical compositions are shown in Table 3 and the results are presented in Figure 6. The first simulations are performed for ERBO alloys where both samples are crept under 160 MPa at 1050 °C for <001> crystal orientation. The $\gamma^{\prime }\;$phase of these alloys consists of cuboidal particles with 72.7% volume fraction and the average size of 500 nm for ERBO-1 whereas the volume fraction and the average size for ERBO-15-MO are 67.1% and 306 nm, respectively, [38]. ERBO-1 alloy that contains more Co and less Mo displays weaker responses as reported in [38] and demonstrated in Figure 6a. Here, the diversion between the simulation and experimental data is mostly related to the rafting phenomenon expected for high temperature which is discussed in the next section. The second simulations are for PWA-1484 alloys for two <001> and <111> orientations at 927 °C under 345 MPa [39]. The morphology of $\gamma^{\prime }\;$for PWA-1484 has a volume fraction of 62.5% and the average size of 300 nm in the shape of cubic precipitates [40]. Once more, the results in <111> orientation are almost linear as shown in Figure 6b. For both cases, the applied load has a low intensity indicating the dominance of climb and glide dislocation mechanisms. The last data set from TMS-238 is tested under a high applied load of 735 MPa at 800 °C with 75% of $\gamma^{\prime }\;$volume fraction and the average size of 250 nm with cuboidal particles [41]. Figure 6c demonstrates strong creep responses due to both composition and morphology of the microstructure. The TMS-238 alloy has a high content of Re and W which are slow diffusers; thus, reduce the effective diffusion coefficient considerably which result in less plastic deformation as shown in Equation (Equation 21). In addition, the $\gamma^{\prime }\;$phase contains the larger volume fraction of precipitates with smaller particles that again causes stronger creep responses which will be discussed in the next section.

#### 3.2.2. Morphology Effects on Creep Curves

The two-phase γ-$\gamma^{\prime }\;$is characterized by three morphology attributes including the shape, average size, and volume fraction of the $\gamma^{\prime }\;$phase. These features are manifested through *n* as the shape factor of the $\gamma^{\prime }\;$phase considered as a super-ellipsoid in the principal directions of $a=b=c=a_{\gamma^{\prime }}$ in the equation ${\left(\frac{x}{a}\right)}^{n}+{\left(\frac{y}{b}\right)}^{n}+{\left(\frac{z}{c}\right)}^{n}=1$. $a_{\gamma^{\prime }}$ specifies the average size of the precipitates and $V_{p}=\frac{V_{\gamma^{\prime }}}{V}$ the volume fraction of the $\gamma^{\prime }\;$phase. Thereby, a broad spectrum of precipitates from spherical (n=2) to nearly cubic appearance (n→∞) can be constructed. For the shape factor, $n_{1}=\tan^{-1}\left(n\right)$ is utilized instead of *n* in order to avoid singularities in cuboidal precipitates. Simulations are performed for single-crystal nickel-based superalloys with the composition of CMSX-4 and the cubic shape of precipitates with the size of 400 nm and 68% volume fraction. All samples are loaded under 400 MPa at 850 °C. To probe the effect of the precipitates shapes, the edges are considered ranging from smooth, semi-spherical with $n_{1}=1.5$, to sharp, roughly cuboidal with $n_{1}=1000$. The results are exhibited in Figure 7a. Climb and glide dislocation mechanisms are dominant at high temperature and low applied loads; thus, imaginably, dislocations will have a difficult time climbing along the sharp edges and glide through the cubic shape precipitates rather than a round or spherical shape for the same size and volume fraction. Consequently, the accumulation of plastic strain will be slower in the case of the former as the plots signify. The second morphology parameter reveals that smaller particles will present a stronger creep behavior. A fixed volume fraction for the same shape of precipitates implies a smaller average size with more particles that will generate more barriers for gliding dislocations. Hence, by reducing the average size of the microstructure, the less plastic strain will be observed with respect to time as set forth in Figure 7b. Distinctly, the volume fraction is the predominantly effective morphology parameter as when increased for the same size and shape of the precipitates, the accumulation of plastic strain remarkably diminishes. In fact, by increasing the volume fraction from 40% to 80% the plastic strain decreases from 10% to less than 1% after 1000 h. Figure 7c, demonstrates that a lower volume fraction leads to easing up dislocations motion; hence, higher plastic strain.

#### 3.2.3. Temperature Effects on Creep Curves—Low to Medium Load Intensities

The dominant dislocation mechanism under low to medium load intensities is climb along the precipitates and glide through the channel where the dislocations do not possess adequate energy to cut through the $\gamma^{\prime }\;$phase. Illustrating the temperature sensitivity of the model, three temperatures of 850 °C, 900 °C, and 950 °C are considered for which the outcomes compared against experimental results in [42]. All tests are carried out on single crystals of CMSX-4 with the composition listed in Table 1 in <001> orientation while fast tensile loading conditions are exerted. Precipitates are assumed to have a cuboidal shape with an average size of 400 nm with a 68% volume fraction. The findings are exhibited in the forms of plastic strain versus time in Figure 8. It is evident that by elevating temperature, plastic stains evolve abruptly in the microstructure and; therefore, decreases the service life. The tests are executed for low to medium load intensities in tension at 850 °C, microstructures experience less than 5% plastic strain after 1000 h for the applied load less than 450 MPa. When the applied load is greater than 600 MPa, the plastic strain in the structure exceeds more that 10% in less than 200 h. As the temperature is increased by 50 °C to 900 °C for an applied load around 370 MPa, the microstructure undergoes more than 10% strain after 1000 h while the plastic strain is less than 1% at 850 °C, for the similarly applied load, the plastic strain accumulates more than 10% in less than 200 h at 950 °C which reveal the evolution of the plastic strain significantly increases by raising the temperature. Here, the slight divergence between the experimental data and the model predictions could be due to the rafting process which is commonly expected for temperature above ⪆900 °C.

#### 3.2.4. Crystal Orientation Effects on Creep Curves—Low to High Load Intensities

To demonstrate the ability of the constitutive model to capture the crystal orientation in both tension and compression loading conditions, two sets of simulations are executed as displayed in Figure 9. The first group is associated with the low to medium load intensities in Figure 9a where all microstructures have the composition of CMSX-4 with 68% of precipitates volume fraction and the average size of 400 nm in the shape of cubes under 850 °C. The crystal orientation is considered close to <001> and <111> [36]. All crystal orientations of <001> indicate exponential plastic strain growth with time while for <111> ones the trend is rather linear. For the high load intensity where dislocations have enough energy to cut through the precipitates, results are compared with experimental data in [43] where clearly specify the asymmetry in tension and compression responses at higher applied loads as illustrated in Figure 9b. The specimen that experiences 820 MPa, almost instantly reaches 10% plastic strain while the sample under 750 MPa, at the same temperature, sustains less than 7% strain after 500 h. In addition, two compression tests at lower applied loads where the climb and glide are the dominant mechanisms are conducted at 950 °C under 250 MPa and 350 MPa. At 350 MPa for crystal orientation of <256>, where both octahedral and cube slip systems are activated, 10% plastic strain occurs in less than 100 h.

## 4. Conclusions

A multi-scale approach to model single crystals of nickel-based superalloys was presented in the crystal plasticity finite element framework which bridges two spatial scales. A detailed dislocation density-based constitutive model was developed for the lower scale with the advantage of explicit representation of the precipitates; hence, higher level of accuracy in the computational outcomes which naturally accompanied by notable processing time. To expedite the computational procedure, the homogenized scale was designed to implicitly incorporate the morphology and composition effects. The constitutive model in this scale demonstrated reliable results for both low and high values of the applied loads where two dislocation mechanisms could occur. At high load intensities, shearing of the precipitates was observed while at lower values and high temperatures, climb along the precipitates and glide through the channel were noted. The model was tested for diverse generations of single-crystals of nickel superalloys and indicated that the slow diffuser elements such as Re and W could reduce the accumulation of plastic strain with respect to time and strengthen creep properties of the alloys. The constitutive model could accurately capture the morphology parameters of the shape, average size, and volume fraction of $\gamma^{\prime }\;$phase whose associated sensitivities were analyzed and assessed. The simulations signified microstructures with sharper, smaller average size, and high volume fraction particles are more suited for creep responses. As a part of future work, the entire configuration is also undertaken with a view to eventually incorporate it into the object-oriented finite element (OOF) developed by the scientists at the National Institute of Standards and Technology (NIST) which is a publicly accessible platform intended to assist materials scientists and engineers in taking part in computational investigations of structure-property relations in a large variety of systems, including mechanical systems whose behavior is dominated by crystal plasticity.

## Figures and Tables

**Figure 1 materials-15-04447-f001:**
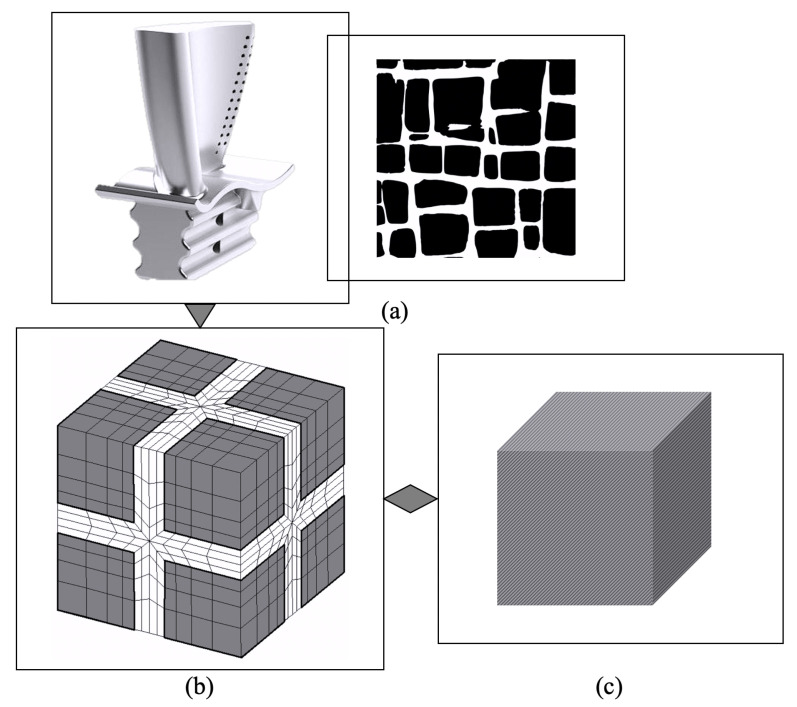
Multi-scale framework of single-crystal nickel-based superalloys: (**a**) Turbine blades with single-crystalline microstructure and cubic precipitates. (**b**) Discretized finite element mesh of a two-phase single-crystal description. (**c**) Homogenized finite element model.

**Figure 2 materials-15-04447-f002:**
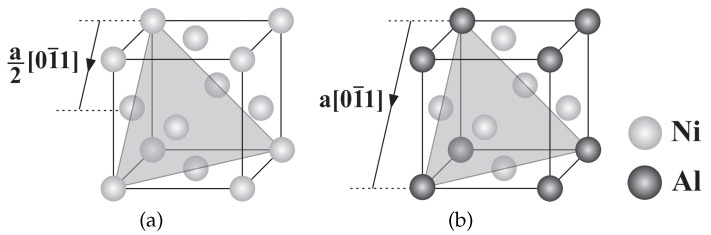
Dislocations in two-phase nickel-based superalloys: (**a**) Crystalline structure of γphase and (**b**) Crystalline structure of $\gamma^{\prime }\;$phase.

**Figure 3 materials-15-04447-f003:**
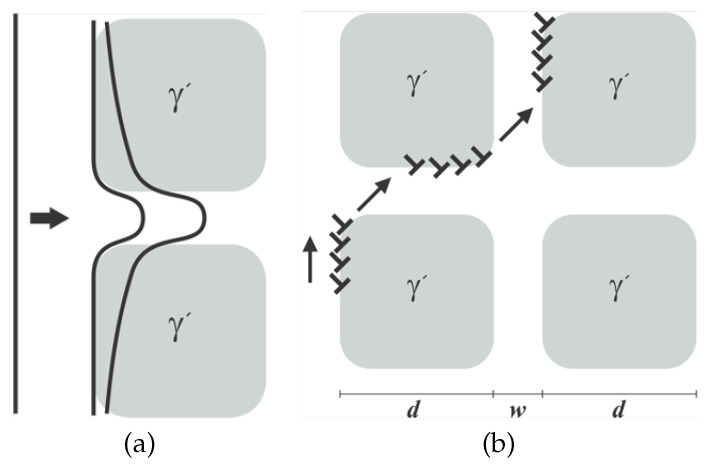
Dominant dislocation mechanisms in nickel-based superalloys: (**a**) Shearing $\gamma^{\prime }\;$and (**b**) Climb and glide.

**Figure 4 materials-15-04447-f004:**
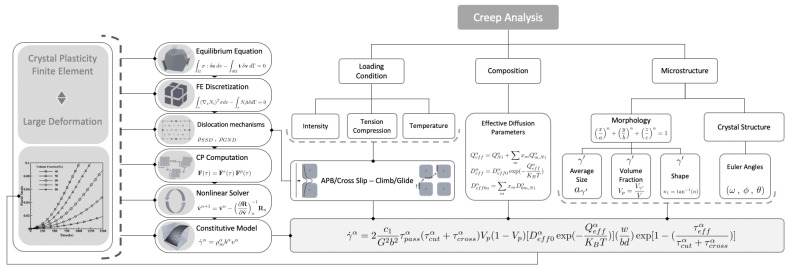
Computational steps to capture creep responses in the CPFE framework.

**Figure 5 materials-15-04447-f005:**
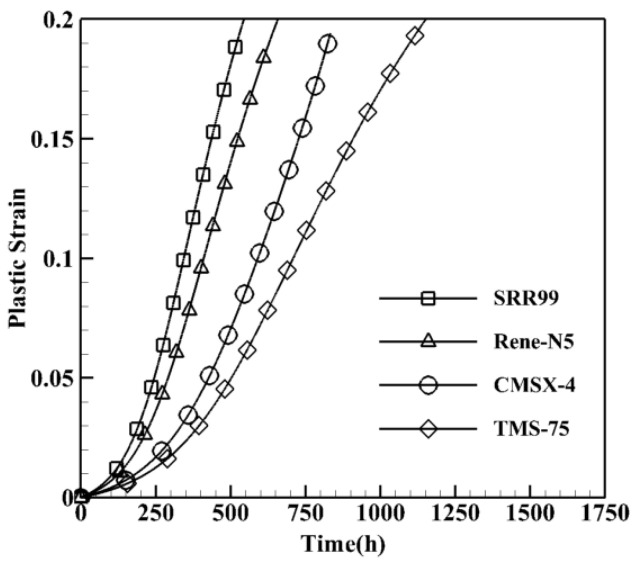
Creep curves obtained by arithmetic method in achieving the effective diffusion coefficient, ($D_{eff0}^{\alpha }$).

**Figure 6 materials-15-04447-f006:**
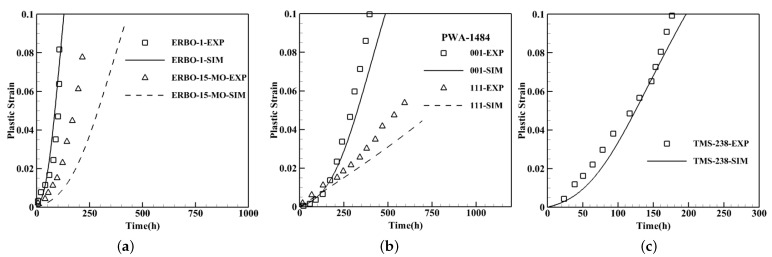
Creep curves for single-crystal nickel-base superalloys with different compositions compared to the experimental data. (**a**) ERBO-1 and ERBO-15-MO [38] under 160 MPa at 1050 °C. (**b**) PWA-1484 [39] under 345 MPa at 927 °C. (**c**) TMS-238 [41] under 735 MPa at 800 °C.

**Figure 7 materials-15-04447-f007:**
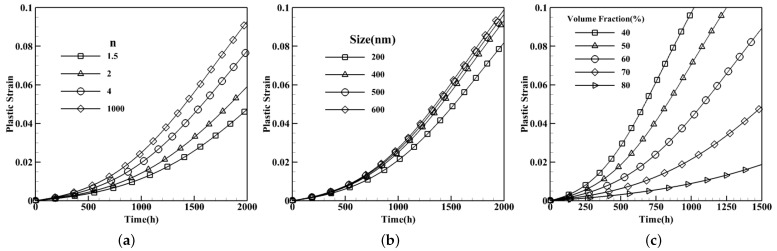
Creep curves for single crystals of CMSX-4 for different morphologies under 400 MPa at 850 °C: Effects of (**a**) shape, (**b**) average size, and (**c**) volume fraction.

**Figure 8 materials-15-04447-f008:**
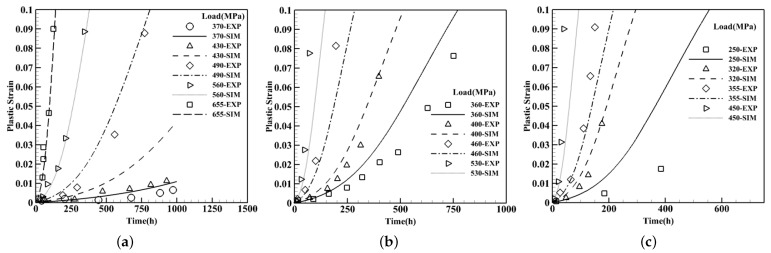
Creep curves of single-crystals of CMSX-4 at different temperatures compared with the experimental data [42] at (**a**) 850 °C, (**b**) 900 °C, and (**c**) 950 °C.

**Figure 9 materials-15-04447-f009:**
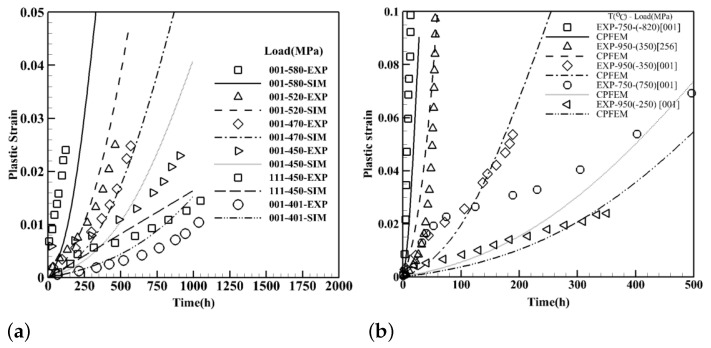
Creep curves for single crystals of CMSX-4 identifying the orientation dependence of the model in tension and compression loading conditions along with the comparison to the experimental data for (**a**) low to medium [36] and (**b**) medium to high load intensities [43].

**Table 1 materials-15-04447-t001:** Nominal chemical composition (mass fraction) for four generations of single-crystal nickel-based superalloys.

Alloy	Cr	Co	Mo	Re	W	Al	Ti	Ta	Ni
SRR99	0.08	0.05	-	-	0.10	0.055	0.022	0.03	Bal
RENE-N5	0.07	0.075	0.015	0.03	0.05	0.062	-	0.06	Bal
CMSX-4	0.065	0.09	0.006	0.03	0.06	0.056	0.01	0.065	Bal
TMS-75	0.03	0.12	0.02	0.05	0.06	0.06	-	0.06	Bal

**Table 2 materials-15-04447-t002:** Pre-exponential constants and activation energies for solute diffusion in nickel.

Element	Cr	Co	Mo	Re	W	Al	Ti	Ta	Ni
$D_{0}\left[m^{2}s^{-1}\right]\times 10^{-5}$	52	7.5	11.5	0.082	1.15	10	14	1.29	19
Q$\left[\mathrm{kJ}\;{\mathrm{mol}}^{-1}\right]$	289	271.7	281.3	255	303	260	263.8	257.5	284

**Table 3 materials-15-04447-t003:** Nominal chemical composition (mass fraction) for four generations of single-crystal nickel-based superalloys.

Alloy	Cr	Co	Mo	Re	Ru	W	Al	Ti	Ta	Hf	Ni
ERBO-1	0.075	0.10	0.004	0.01	-	0.021	0.126	0.013	0.022	0.0003	Bal
ERBO-15-MO	0.076	0.031	0.013	-	-	0.025	0.113	0.04	-	-	Bal
PWA-1484	0.05	0.10	0.02	0.03	-	0.06	0.056	-	0.09	0.001	Bal
TMS-238	0.046	0.065	0.011	0.064	0.05	0.04	0.059	-	0.076	0.0002	Bal

## Data Availability

The data that support the findings of this study can be accommodated upon reasonable request.

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
