# Peer review of "Multi-Scale Crystal Plasticity Model of Creep Responses in Nickel-Based Superalloys"

_materials, 2022, doi:10.3390/ma15134447_

Round 1
Reviewer 1 Report
The work of the authors present an interesting multiscale computational framework on the mechanical behaviour of single crystal superalloys under different temperature and stress conditions, field in which the authors present a noticeable background. The model captures succesfully dependences on the chemistry and microstructural features. The works is carefully designed and the writting is well cared. However, there are some important points that the authors might consider:
-The authors have a good background in the field of computational modelling of Ni-based superalloys. However, the manuscrupt is missing a literature review section of the differerent computational approches already published in the past and how the new work presented in the manuscript innovates over them.
-The manuscript would benefit from references to metallurgical facts like in the part where the authors explain the different deformation mechanisms present in a superalloys in Page 3
-Reference is missing in Page 7 where the v is introduced (before equation 2.4)
-Please be consistent with the variable names (b_alpha in Orowan equation but only b in 2.6 and after)
-Please introduce/explain all the parameters you are using, for examples c2, c3,c9 (also, is c9 a typo and should be c4? I am missing c5-c8 otherwise)
-At the low scale model, there is no climbing, but climbing is included at the high scale level. Can the authors please explain/justify this aspect in the paper to assure the consistency of the mechanisms accross scales?
-The dependency on the vectorial form of GNDs in equations 2.7 is not clear. Only SSDs variables are present in that equations (rho_F and rho_P) Can the authors explain?
-Page 8 end, This process creates a planar fault,... or APB "within the Gamma prime".
-Page 9 and 10: currently the nomenclature for gliding dislocations is confussing (rho). As you also include mobile and climbing dislocations (rho_m and rho_c), I suggest that the authors could name gliding dislocations as rho_g
-The manuscript will benefit from a table of calibrated constants and their values for all the alloys (e.g., c1,c2,c3,c9) so the study is replicable.
Also, please check the following typos:
-Page 3 "fproperties"
- Page 8 "scaler"
Reviewer 2 Report
It is a very interesting article on γ' strengthened superalloys.
1. If the authors considered other - microstructural precipitates? Right now, it is focused on gamma and gamma prime phases only. If so, e.g., in the case of TCP phases? is there a way to capture elemental segregation effects on the creep model & mechanism at different temperatures?
2. It will be good to comment on the applicability to other single / poly-crystalline Nickel superalloys.
But they are just seeking clarifications and the paper looks good to go except for any typos and minor edits.
Round 2
Reviewer 1 Report
The manuscript has improved in the this new version considerably and from the point of view of the reviewer is ready to be published except for a minor points:
-There is still b (burguer vector symbol) without an alpha